# Emotional Intelligence and Critical Thinking: Relevant Factors for Training Future Teachers in a Chilean Pedagogy Program

**DOI:** 10.3390/jintelligence13020017

**Published:** 2025-01-31

**Authors:** Maritza Palma-Luengo, Nelly Lagos-San Martin, Carlos Ossa-Cornejo

**Affiliations:** Educational Science Department, University of Bío-Bío, Chillán 3780000, Chile; mpalma@ubiobio.cl (M.P.-L.); nlagos@ubiobio.cl (N.L.-S.M.)

**Keywords:** critical thinking, emotional intelligence, professional training, pedagogy

## Abstract

Critical thinking has become one of the most notable cognitive skills in education in recent decades since it offers skills for improving knowledge, making decisions, and creativity, among others. While it is considered a mainly cognitive process, recent years have seen strong proposals regarding its relationship with motivational and emotional processes. A study is presented that analyzes the relationship between critical thinking and emotional intelligence, analyzing the relations and influences between these variables. Two instruments were applied to 658 Chilean pedagogy students with ages ranging from 19 to 47 years old. The results indicate a moderate level of critical thinking and a high level of emotional intelligence, along with a positive and significant but moderately low relationship between emotional intelligence and critical thinking. There are no gender differences, meaning that men and women developed these skills in a similar way, but differences were found between age groups. The study highlights the relevance of promoting both critical thinking and emotional intelligence in training future teachers and the need to generate new studies about how these skills are developed in teacher training.

## 1. Introduction

Emotional intelligence (EI) and critical thinking (CP) are two essential skills in the training of future teachers since they directly impact their ability to face the challenges of the educational environment and develop effective relationships with students. Thinking skills are fundamental in the educational field, and their improvement has become a key objective in teaching systems ([16]). Through various methods, such as critical thinking, the aim is to enhance the cognitive and emotional development of individuals, a process that, according to [92] ([92]), is crucial to integrating emotional intelligence into daily life.

The development of critical thinking in university students has been a topic of growing interest. Studies appear from the students’ conception of critical thinking versus that of the teachers ([5]) to others related to the importance of integrating critical thinking skills into the university curriculum, using active methodologies and interdisciplinary approaches ([18]). It has been observed that the implementation of disciplinary reading and writing practices, as well as the use of socio-scientific dilemmas, have been effective in strengthening these skills ([55]). Furthermore, studies have shown that students who participate in research activities and collaborative projects tend to develop more robust critical thinking ([72]).

EI is the ability to recognize, understand, and regulate emotions to promote intellectual and personal growth ([82]) and plays a determining role in the academic and personal success of individuals. In fact, previous research has shown that EI levels directly influence non-cognitive beliefs, skills, and abilities, factors that are essential to facing everyday challenges ([70]; [43]; [69]; [52]). In this sense, for teachers in training, developing both their EI and PC not only improves their academic performance but also allows them to better manage their emotions and make reflective decisions that favor the comprehensive development of their students ([7]; [43]).

On the other hand, research points to a close and binding relationship between critical thinking and some emotional aspects related to motivation ([14]). The promotion of motivational aspects is a relevant factor for the development of cognitive and metacognitive processes since complex processes are exhausting and require a high and constant investment of cognitive and emotional factors ([32]).

Although emotional intelligence and critical thinking are distinct skills, there is debate over whether they are redundant constructs due to their interrelationship and overlap with other cognitive abilities and personality traits ([23]; [44]). For the same reason, the relationship between critical thinking and emotional intelligence has been the subject of great controversy and debate in the analysis of various studies. Mainly because some authors make the separation between the emotional and the rational, generating a common criticism, which is the idea that critical thinking is based on analysis and objective evaluation, and, therefore, should not be influenced by emotions. For [91] ([91]), if memory depended entirely on emotions, we would remember very little of what we experienced in the educational system. For him, it is more accurate to say that things that create emotional reactions will be remembered better, but emotion is not a necessary condition for learning. On the other hand, some authors argue that emotions can hinder judgment and lead to biased decisions ([92]). This criticism refers to the difficulty in emotional self-regulation, focusing on the difficulty that many people have in regulating their emotions effectively ([16]). Lack of self-regulation could interfere with the ability to think critically since stress and anxiety would negatively affect decision-making; Related to that, a negative impact on cognitive effort has also been noted, referring to the negative correlation between certain aspects of emotional intelligence and critical thinking. For example, greater emotional clarity may be associated with lower cognitive effort, which could reduce the depth of critical analysis ([28]).

The incorporation of critical thinking skills can enhance the training process in Chilean Initial Teacher Training, an aspect that urgently needs to be worked on to provide education professionals with better tools to face current challenges ([2]). In some countries, there are critical thinking development programs aimed at certain areas related to education, predominating in English-speaking countries more than in Latin American countries ([62]).

Both processes are at the core of the university’s educational policies in Chile, related to 21st century skills; but, the relationship between both concepts is underdeveloped in the Chilean context. With a few studies researching these links, this study aims to analyze the relationship between critical thinking and emotional intelligence in Chilean university teacher training students, considering sex and age characteristics by the lack of research in this area.

### 1.1. Emotional Intelligence

Emotional intelligence (EI) has become an essential component for training teachers, as it significantly influences the quality of teaching and the well-being of both students and teachers alike ([8]; [66]), given that EI refers to the ability to recognize, understand, and manage emotions within oneself as well as with others ([33]; [50]; [80]).

While emotional intelligence has been widely valued as a tool in education and psychology, it has also been heavily criticized in aspects such as its purpose, its ability to be defined, and its measurement ([10]; [56]). However, the widespread use of the concept shows that, although it requires further specification and objectification, it has been useful in understanding the social and emotional sphere.

Including EI within teacher training helps develop crucial skills for emotional management in the classroom, which can help improve the academic climate and relations with students ([36]; [40]). This element highlights the importance of teachers having a more polished knowledge of their own emotionality and the ability to regulate their emotions in order to then educate and emotionally orient their students via internalized tools ([17]).

Likewise, David [86] ([86]), in his book “A Nation at Thought” calls for reforming the American educational system, focusing on the development of a comprehensive education, which includes new ways to use our thoughts for effective learning and values social emotional learning in students so that to be able to reach their full potential and contribute positively to society. Although emotional skills are difficult to measure and work on, their relationship with well-being and discomfort behaviors means that they can be observed and worked on as indirect measures, especially through scales and tests ([58]).

Regarding the most commonly used instruments for measuring EI in professionals since 2020, a systematic review by [10] ([10]) indicated that the most frequently reported tools were the Emotional Quotient Inventory (EQ-i), the Schutte Self-Report Inventory (SSRI), the Mayer–Salovey–Caruso Emotional Intelligence Test 2.0 (MSCEIT 2.0), the Trait Meta-Mood scale (TMMS-24), the Wong–Law Emotional Intelligence Scale (WLEIS), and the Trait Emotional Intelligence Questionnaire (TEIQue). These authors also stated that the aforementioned instruments had adequate psychometric properties for confidentiality and validity, meaning that studies whose results were obtained based on these instruments are more trustworthy.

EI has been studied in relation to other variables, such as empathy ([20]; [4]) and interpersonal relations ([78]). In both cases, this association is important since they are used to show that EI has a social component that makes it possible to achieve successful social interaction ([30]). In turn, EI helps teachers manage stress and prevent professional burnout, promoting a more positive and effective environment for learning and teaching ([51]). This fact also increases the relevance of EI due to teachers’ high daily stress levels.

The relevant empirical results include the relationship between EI and future teachers’ self-esteem ([83]), demonstrating a positive relation between emotional intelligence and self-esteem, concluding that the teacher training program should revamp its curricula via designing intervention strategies to improve future teachers’ EI and self-esteem. Research has also been done about EI and learning results (social, cognitive, personal growth, and satisfaction with the university experience) among students at Chinese universities ([81]). The study results reveal that EI has a significant impact on learning results. An indirect relation is also established between EI and learning results via students’ trust in teachers and in the orientation of the learning. A direct relation is also established between learning results and students’ academic efficacy. In turn, [13] ([13]) referred to EI among principals and their teaching leadership behavior, which are influential factors for teachers’ teaching strategies. These findings are particularly interesting because they include EI for the improvement of teaching praxis and evaluating principals’ effectiveness. Finally, [54] ([54]) analyzed the effect of the formal and informal emotional education received when acquiring emotional skills among teachers of Preschool and Primary Education, who showed a significant influence from the training received by teachers when acquiring emotional repair skills, one of the core dimensions in the instrument which was used (TMMS-24), regarding the use of emotional self-regulation strategies and regulation of more complex psychological processes, implying the application of other emotional skills.

### 1.2. Critical Thinking

Even though critical thinking is a widely used concept in the academic and educational world, with a large amount of research in the last decade, it continues to be a phenomenon that is difficult to conceptualize and with little consensus among researchers and trainers ([19]; [74]; [62]). We can indicate that critical thinking is a process oriented toward analyzing and effectively using information linked with problem-solving and decision-making. This type of thinking is made up of specific skills that report data, help evaluate it, and place it in the service of a situation that must be resolved ([59]; [77]). In other words, it is a way to use thought in order to analyze, decide between alternatives, and evaluate this decision ([75]).

The literature also indicates that critical thinking (CT) is a process that can be developed in a sociocultural space and can be systematically and progressively molded because the skills included within it are complex ([74]) and correspond to a type of elaborated thought; i.e., a cognitive process involving evaluation and reflection ([11]). It is considered a type of complex cognitive process, composed of interrelated subprocesses, making it possible to evaluate, analytically, and reflectively process, judge, and accept or reject information produced within social contexts or in scientific studies ([88]).

In this order, CT can be considered by certain perspectives like a fuzzy idea, difficult to apply in schools’ realities ([86]); However, despite this inherent ambiguity, based on the fact that thought is something immaterial, it has been possible to conceptualize and objectify it based on visible thinking behaviors ([71]), which allow its empirical observation and evaluation ([75]).

This type of thinking allows for the construction of new knowledge, and the strategic application of this knowledge in solving problems arising in everyday life ([9]; [48]). Its acquisition requires intentional programming, and this skill is oriented toward information and action within a problem-solving context and during interaction with other people ([62]; [74], [75]). Its goal is to generate reflective judgment based on developing cognitive skills, translating into the ability to resolve situations happening inside and outside the classroom ([48]).

The most relevant thing about this construct is its multifaceted character since it includes various cognitive processes including analysis, evaluation, inference, and interpretation, as well as because it implies the ability to question assumptions, examine evidence, and consider alternative perspectives, leading to well-reasoned conclusions and informed decisions ([65]). There are many CT tests that have been used to measure and promote critical thinking, some of them, based on an argument reading; others based on situations of use of logic through alternatives; a third group uses cases on everyday life situations to apply cognitive processes related to CT ([75]).

Currently, there are many studies that show how critical thinking has been incorporated into higher education institutions around the world, providing information in areas of skill measurement, relationship with other soft skills necessary for the 21st century, strengthening programs for the scientific field, and as support to strengthen reading and writing skills, among the most named ([22]; [63]; [57]). Within educational contexts, critical thinking is cultivated via activities encouraging active participation in the content, such as problem-based learning, case studies, and collaborative debates ([27]). This helps identify a set of basic elements comprising critical thinking, with cognitive and metacognitive elements standing out, such as reasoning, problem-solving, decision-making, and metacognition ([76]; [73]), along with others of a motivational and emotional nature, which presents a disposition towards the tasks carried out by this cognitive process, i.e., of an interest which promotes the use of knowledge and decision processes ([59]; [77]; [89]).

Specifically in educational careers, there are various experiences for the development of critical thinking in pedagogy students in Latin American countries; however, the number of them would be low compared to the experiences carried out in other continents and in other disciplinary areas, possibly for political and epistemological reasons; In Chile, these studies are even less developed ([60]).

### 1.3. Relation Between Emotional Intelligence and Critical Thinking

Critical thinking has traditionally been linked with intelligence, understood as a cognitive process and an effective decision-making process, since both are complex information development processes ([68]). However, the mere development of intelligence as an analytical capacity is not always effective in adjusting to the real world, given that this capacity is often only related to mechanical or memory-centric data processing and not to the complexities of making decisions in everyday life ([34]).

Critical thinking could be a central axis in the idea of intelligence when extended beyond effective data analysis into the consequences of making decisions, along with the impact of fallacies and biases that accompany information in sociocultural media. This requires not only cognitive processing but also motivational and reflective processing ([34]). In this way, critical thinking considers emotional–motivational aspects that affect reasoning and decision-making ([37]; [61]).

Some research indicates that the relationship between EI and PC is more complex and less direct than some theoretical models propose, depending on variables such as the cultural, educational, and methodological context. A study with nursing students concluded that the general correlations between EI and CP were not significant, but identified that some specific emotional aspects, such as empathy, have a limited relationship with subcomponents of CP, which highlights a non-uniform connection ([35]). [46] ([46]), in a meta-analytic analysis, pointed out that although EI is associated with general academic skills, the relationship with critical cognitive competencies, such as PC, can be moderated by variables such as cultural context and the specific definition of both skills.

Despite these criticisms, some experts also highlight that emotional intelligence and critical thinking can complement and strengthen each other. The key is to find an appropriate balance and develop both skills in an integrated way ([70]; [84]; [82]). This comprehensive approach not only benefits academic performance but also contributes to the well-being and personal development of everyone involved ([43]).

According to [69] ([69]), EI allows teachers to face work stress and create positive learning environments that promote both the academic performance and emotional well-being of students. It has also been shown that nurses with higher levels of emotional intelligence also scored higher in commitment, being the interpersonal factor with the greatest prediction ([67]). Meanwhile, PC, which involves the ability to analyze, evaluate, and generate ideas in a logical and reflective manner, is directly related to improving decision-making. Studies such as those by [3] ([3]) demonstrate that integrating critical skills into the educational process promotes the intellectual autonomy of students, enhancing their meaningful learning and their ability to solve complex problems.

That is, the combination of these competencies has a multiplying effect when EI provides the emotional stability necessary to address conflicts or ethical decisions in the classroom and PC allows structuring solutions based on well-founded reasoning. This is also explained by [34] ([34]) through the idea of the complementarity between emotional intelligence as the appropriate and strategic use of emotions and critical thinking as a strategic decision process.

While this topic has seen little research and mostly recent data, with a low number of studies indicating the relation between the two variables, we can find a bibliographical analysis to analyze the state of the art in the relation between EI and CT from [45] ([45]) which indicates that 11 studies were found that analyzed the links between CT and EI, and only 2 of these ([42]) established correlations, which were positive and significant. According to this review, the relationship between EI and disposition toward CT has been studied ([85]), with no significant relation being found, along with a study about the relationship between critical thinking and socio-emotional skills ([38]), which presented a significant relation only in the empathy dimension.

Another bibliographical review referring to training for healthcare professionals found that both variables are interdependent, that both are fundamental for academic success among health professionals, and finally, that professional training should integrate them more explicitly to achieve this positive impact ([15]).

In an empirical study with 471 Turkish tourism majors, [37] ([37]) found a positive and significant correlation between the dimensions of EI and the dimensions for inclination towards CT. However, they mentioned that even when the relation between EI and the motivational dimension of CT is empirically and theoretically feasible, there is insufficient clarity about whether the same relation would exist between EI and the cognitive domains of CT ([37]).

Despite the disparate results for the relation between the variables, [24] ([24]) indicated that theoretically there would be a clear relation between both since emotional intelligence can be considered as a measurement of the extent to which a person can successfully (or unsuccessfully) apply proper judgment and reasoning about situations in the process of determining emotional responses.

Similarly, [21] ([21]) indicated that complex cognitive processes such as metacognition and critical thinking are the means that emotionality requires to move towards a more transcendent and socially adequate development.

## 2. Materials and Methods

The study had a non-experimental design with a correlation descriptive character.

### 2.1. Participants

The participants were 658 university students with eight pedagogy majors in a Chilean higher education institution. Of these, 245 are male (37%) and 413 are female (63%). Ages ranged from 19 to 47 years, with an average age of 21.6 years and a standard deviation of 2.86. These participants were invited to participate in the initial activities of their courses in accordance with the academics who led those courses, using a non-probabilistic and convenience sample.

### 2.2. Instruments

Two instruments were applied. The first of these was the adapted critical thinking tasks questionnaire ([64]), which measures the critical thinking variable based on 15 open and closed questions, evaluated via a rubric with values between 0 and 2 pts. The items are organized into 5 dimensions (inquiry, analysis, argumentative communication, metacognition, and motivation), in a mixed format that includes multiple choice and writing responses, like is shown in Figure 1. Adequate internal consistency values have been reported, with an overall Cronbach’s α between 0.67 and 0.78 ([64]; [61]). 

The second instrument used was the Trait Meta-Mood Scale TMMS-24 ([26]), consisting of 24 items organized into 3 dimensions (attention, clarity, and emotional repair). The scores are organized on a Likert scale from 0 to 4 points. This scale has been validated among Chilean university students with excellent reliability values, achieving a Cronbach’s α of 0.95 ([25]).

As there are no typified values for the Chilean population in both instruments, the scores were transformed into a percentage for a common basis.

### 2.3. Procedure

Students were contacted via university authorities, and invited to one of the classes, to respond to the instruments. The evaluation battery included written informed consent indicating the study objectives, the goals of the survey, anonymity in the responses, and the voluntary nature of participation.

The data gathered was emptied into a database and analyzed using the SPSS v. 24 statistical program. For analysis, we used descriptive statistics (measurements of central trends, dispersion, and distribution), correlation (Pearson’s r), and differences in medians (Student’s t-test, one-way Anova, and Kruskal–Wallis).

### 2.4. Ethical Statement

This project was reviewed and approved for ethical considerations by a bioethical committee of the University of Bío-Bío in 2023.

## 3. Results

The results show that at the descriptive level, the values for the measured variables are above the arithmetical means for the instruments in both variables (Table 1). There is also a greater dispersion of scores in critical thinking than in emotional intelligence. When considering the distribution characteristics, critical thinking presents values indicating a normalized distribution; however, the emotional intelligence variable has high kurtosis, with a platykurtic distribution.

It is also possible to observe a relation that is highly significant, but simultaneously weak between both variables (r = 0.212, *p* < 0.01). This relation is worth consideration, as it might indicate a link between the variables meriting further research.

Regarding the gender differences, Table 2 indicates that there are no statistically significant differences between men and women in emotional intelligence; however, we can note that in critical thinking, even when the value of p is above 0.05 (95%), which is the consensus for establishing statistical significance, it still falls within significance values which could be accepted (90%), although with more possibility of error in rejecting the null hypothesis, since a type II error could occur ([47]; [53]).

To contrast the preceding point, we applied an ANCOVA to determine whether there could effectively be any gender influence in the critical thinking result. However, the data indicate that even when managing the gender covariable, we can observe that its value does not generate an impact on the critical thinking variable (F = 2.37, *p* = 0.144, with r2 = 0.048), thus confirming that there are no significant differences in this variable and that gender does not influence the result.

Finally, an age distribution analysis was done for both variables using three groups, considering the course levels in the institution: 19–20 (first university grade); 21–22 (second university grade); and 23 years and above (third university grade). Table 3 shows students corresponding in all three grades.

The data distribution (Table 4) indicates that there are differences between the groups in the critical thinking variable, but not for emotional intelligence.

The ANOVA analysis indicates a statistically significant difference for CT between age groups (*p* < 0.001), with the 19–20 age group standing out as having a higher mean (73.43) compared to older age groups. Meanwhile, EI presents no significant differences between groups (*p* > 0.05). The Kruskal–Wallis test results also reflect these conclusions.

The preceding points are graphically displayed in Figure 2, which clearly indicates the varying distribution of test values between group 1 and the other groups.

In summary, age does not seem to significantly influence levels of emotional intelligence, although it is slightly significant in relation to critical thinking according to these analyses.

## 4. Discussion

The results offer information about skills related to critical thinking and emotional literacy amongst future teachers, with a greater dispersion of CT which could indicate a broader range of skills among students, along with being beneficial for diversifying educational strategies in the classroom. In turn, the platykurtic distribution of EI suggests that most of the students are clustered around the mean, with few extremes, which indicate that the students have a more homogeneous behavior in this variable and need to strengthen their emotional skills. In turn, this could facilitate the implementation of emotional development programs to approach students’ common needs.

While we can understand that complex processes such as critical thinking as related to intelligence, a high IQ is not always an indicator of good critical judgment, since sometimes, people with high intelligence are not exempt from biases or rigid thoughts ([34]). The same occurs with the relationship between emotional intelligence and cognitive intelligence, finding a weak and even negative relationship between both factors ([31]; [87]). Thus, even considering that CT and EI, even when they share a cognitive foundation, it cannot be assumed that the relationship between both processes is something obvious.

There is a significant, albeit weak, relation between CT and EI, with results similar to those found by [45] ([45]), [38] ([38]), and [41] ([41]), who revealed that the EI level is not a determining factor in the rise in students’ CT skills with regards to learning styles. However, a significant correlation between both variables was found by [1] ([1]), while [16] ([16]) suggested that an alternative and optimal way for educators to improve their students’ CT is through EI instead of with EI, as previously believed.

Because no significant gender differences were found for both variables, we suggest that both men and women develop these skills similarly, as also pointed out by studies ([39]). However, it is germane to note that there are few studies about the relation between EI and the disposition towards CT with regard to gender ([84]), indicating a need for more research in order to find more conclusive confirmation for these findings. There is a line of argument indicating that a differentiated response is needed for these two variables, associating aspects of gender roles in the manifestation of EI and CT. However, there is low certainty about whether the study variables are related to gender-based dispositions ([49]), upholding the need to approach them from a non-gender and non-sexist focus ([90]). This implies modifying the definition of current roles, and pedagogy students are beginning to realize this sociocultural transformation.

Concerning the age distribution, the behavior for EI is similar between genders without any significant differences, while age does present differences between groups, with more development in younger students (19 and 20 years old) than for those 21 and over. These data lead us to think that there was a superficial contradiction, given that we expect older people to have higher levels of critical thinking than younger people. However, there are few studies that clearly establish this relation ([6]).

A variety of possible causes can be put forward, on the one hand, that there are difficulties in understanding the questions in some of the age groups, or that there is greater demotivation in the older age group due to the fact that they are faced with greater training demands. However, we can also hypothetically propose a possible reason based on the sociocultural characteristics associated with the new generations’ sociocultural and values-related makeup, given that they come from the so-called millennial and centennial generations. Young millennials are people who were born into a world with significant cultural activity arising from major social changes and technological development, corresponding to people born between the 1980s and 2000, although some authors place the cutoff around the year 2003/2004 ([12]). These are the students who are currently 21 years old or more, and while their ideology is fairly flexible and postmodernist, it still contains elements of modernism, even valuing certain characteristics of the establishment more if their own financial tranquility depends on it ([93]). Centennials are young people born after the 2003/2004 period, a period of technological and digital contextualization characterized by wars, corporate monopolies, and socioeconomic competitiveness. However, they are more conscious of the importance of ecology, democracy, and personal freedom ([12]).

Young centennials are in line with postmodern values and have lived in a world that values participation, human rights, and personal well-being, with less monetary inclination and stronger critiques of the current world ([12]; [93]). In this way, we could hypothesize that young people in group 1, who were 19 and 20 years old, would fall within the group with centennial ideology, being more propositive and critical, and thus with higher levels of critical arguing, while young people in groups 2 and 3 who were 21 and over would be in the group more related to millennial ideology, less critical of situations or less inclined to question and give their personal opinions.

Teacher training programs must, therefore, integrate strategies that simultaneously promote critical thinking and emotional intelligence since both skills are needed for a modern professional whose primary formation should improve academic performance ([46]) and adapt better to the changes and challenges of a globalized world. From another perspective, it is relevant to highlight the findings from [16] ([16]), indicating that EI can be the underlying mechanism to achieve CT, which must be applied and adequately cultivated in a learning environment, thereby suggesting that universities could modify their study plans and place emotional intelligence at the epicenter of teaching.

More specifically, to improve critical thinking and emotional intelligence in student teachers, it is essential to integrate practical and reflective strategies. Regarding critical thinking, the use of debates, case analysis, and open questions fosters logical reasoning and structured argumentation skills, allowing students to explore diverse perspectives ([29]). Furthermore, critical reading and reflective writing activities facilitate the evaluation of arguments and the development of deeper and more autonomous thinking ([1]). On the other hand, emotional intelligence can be strengthened through emotional self-regulation workshops that include mindfulness techniques, as well as dynamics to promote empathy and recognition of the emotions of others ([35]; [79]). The implementation of these strategies within the pedagogical curriculum could improve both academic performance and the ability to manage emotions and interpersonal relationships effectively in educational contexts.

Active methodologies are also very important, such as problem-based learning (PBL), Phenomenon-Based Learning, and Design Base Learning (DBL), they are key tools to promote emotional intelligence and critical thinking in students. These strategies promote active participation, reflective analysis, and problem-solving in real contexts, strengthening skills such as emotional self-regulation and the ability to argue in a reasoned manner ([1]; [29]). Furthermore, they integrate dynamics that enhance empathy and effective communication, pillars of emotional intelligence, while stimulating logical reasoning and decision-making ([35]; [79]). Its implementation in educational environments favors meaningful learning and the comprehensive development of the student, essential for their professional and personal performance.

## 5. Conclusions

The results from this study reveal disparity in the critical thinking and emotional intelligence skills of future teachers, highlighting the need for educational diversification and emotional reinforcement, due to the lack of significant gender differences and the variability in the relation between EI and CT. This highlights the complexity of these skills and the need for non-sexist, personalized focuses in teacher training.

Although it has been suggested that CT is conditioned by sociocultural characteristics and that socioeconomic level can influence the use of cognitive skills ([94]), there is a growing social movement that seeks to grant greater freedom and participation to citizens, and in this, critical thinking is fundamental; this is reflected in the difference between the age groups found regarding critical thinking in the participants.

The study data can also contribute to the discussion about whether the variables present gender differences, indicating that in this group of participants, no differences were observed between men and women for EI and CT. Contributions can also be made to scientific discussions regarding possible differences in the development of CT due to participants’ ages, which could reflect sociocultural influences in this variable, suggesting that the centennial generation could be better prepared to face contemporary challenges.

These conclusions support the integration of strategies simultaneously promoting CT and EI in teacher training programs, adapting them to the demands of a constantly changing and globalized world, as recent studies propose. In this way, the balance between these skills is essential for training future educators, who must be able to encourage learning environments that are inclusive, reflective, and emotionally intelligent.

Concerning the limitations of the study, we must consider that while it is true that the analysis performed shows a significant relation between EI and CT, this relation is weak, meaning that it would be convenient to expand the sample with a larger student body in order to see whether the results remain significant, and how these variables are related in a larger sample. It would also be necessary to include a longitudinal study to see whether the emotional and CT skills learned can be maintained or expanded over time across training, in order to show the effects of achieving exit profiles and the impact on professional teaching praxis.

The findings may not be applicable to other disciplines or social contexts outside of the pedagogy programs studied at a Chilean university. For this reason, it is recommended to investigate the observed patterns of both variables in other undergraduate courses and in other cultural contexts.

Although it has been confirmed that the sample used for the study highlights the importance of encouraging both CT and EI when training future teachers, there is also a need to generate new questions about how these skills develop and are interrelated, since there is a need for an approach to these two constructs which is empirical and not only theoretical in order to move towards more integral development of professional skills.

## Figures and Tables

**Figure 1 jintelligence-13-00017-f001:**
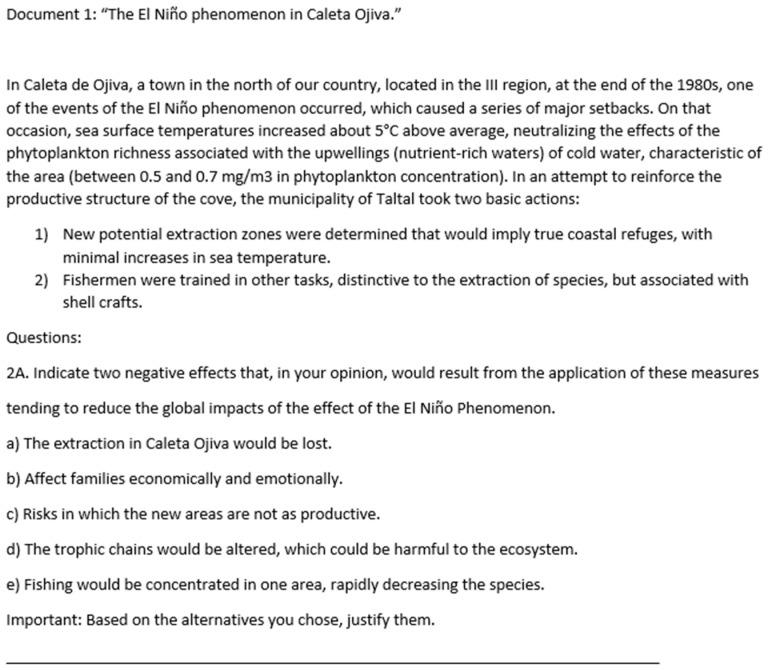
Type of question included in adapted critical thinking task questionnaire.

**Figure 2 jintelligence-13-00017-f002:**
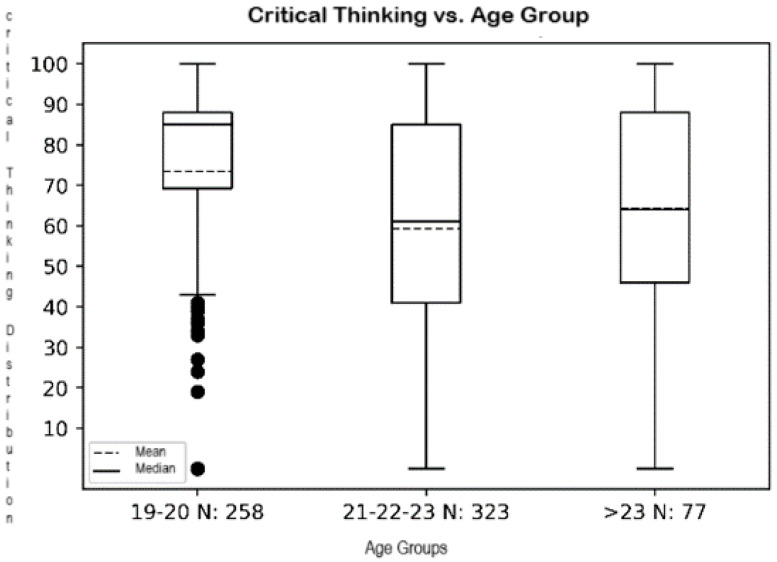
Graph representing the variables for age groups and Critical Thinking.

**Table 1 jintelligence-13-00017-t001:** Descriptive values of the variables.

Variable	Min.	Max.	Median	S. D.	Asymmetry	Kurtosis
Emotional Intelligence	0	100	67.96	19.54	−1.864	4.238
Critical Thinking	0	100	65.43	27.82	−0.986	0.025

**Table 2 jintelligence-13-00017-t002:** Statistical comparison between women and men for Emotional Intelligence and Critical Thinking.

	Mwn ^1^ (DE)	Mmn ^2^ (DE)	*T*	*p*	CI Lower Limit	CI Upper Limit
Emotional intelligence	68.69 (19.26)	66.74 (19.96)	−1.23	0.217	−5.038	1.148
Critical thinking	66.84 (27.67)	63.05 (27.94)	−1.69	0.091	−8.184	0.611

^1^. Mwn = Women Mean; ^2^. Mmn = Men Mean.

**Table 3 jintelligence-13-00017-t003:** Number of participants distributed by age groups.

Variable	Group 1	Group 2	Group 3
Emotional intelligence	258	323	77
Critical thinking	258	323	77

**Table 4 jintelligence-13-00017-t004:** Statistical comparison by age groups for EI and CT.

Variable	Mgroup 1 19–20	Mgroup 2 21–23	Mgroup 3 >23	F ANOVA	*p* ANOVA	F Kruskal-Wallis	*p* Kruskal-Wallis
Emotional intelligence	68.1240	67.4954	69.3896	0.3058	0.7366	0.3058	0.4369
Critical thinking	73.4302	59.3003	64.2987	19.6375	0.0000	19.6375	0.0000

## Data Availability

The dataset is being evaluated in the Zenodo repository.

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
