# Peer review of "Emotional Intelligence and Critical Thinking: Relevant Factors for Training Future Teachers in a Chilean Pedagogy Program"

_jintelligence, 2025, doi:10.3390/jintelligence13020017_

Round 1
Reviewer 1 Report
Comments and Suggestions for Authors
See attachment.

Author Response
Comment 1: First, the authors should add a paragraph or two to grapple with the critics of these two concepts
Response: A paragraph considering critics about this concepts was included
Comment 2: Second, as researchers like E.D. Hirsch and Daniel Willingham point out, critical thinking rests heavily on knowledge, without which we have little to be critical about. This should be explored.
Response: A brief definition about the nature of critical thinking was added, but isn't focus of this paper discusse the foundation of CT or his multiple perspectives.
Comment 3: Third, the definition of critical thinking used here is class bound. As researchers like Annette Lareau (Unequal Childhoods) and more recently Jonathan Haidt (The Righteous Mind) point out, working class child-raising stresses obedience more than questioning. This is not to say that the
concept is invalid, merely that it is class-bound, and this should be discussed.
Response: A paragraph was included in the conclusions about the impact of sociocultural and economics trends in cognitive task.
Comment 4: Finally, here, the two concepts slightly correlate. Of course, if both emotional intelligence and critical theory are related to G (intelligence), the relationships found here could be spurious. This
is my key concern.
Response: a paragraph with ideas about relationships between CT, EQ and IQ was added in discussion. There are studies showing positive correlations between the IQ end the EQ and others, shows a negative correlations. The empirical relationship between CT end IQ is unclear
Reviewer 2 Report
Comments and Suggestions for Authors
Both emotional intelligence and critical thinking are essential skills for success in today’s world, capturing attention in both scholarly and applied domains. Exploring the relationship between these two constructs is particularly significant, especially in the context of higher education. However, this study presents several critical flaws that raise serious concerns.
First, the study lacks a thorough introduction to the current state of research on critical thinking, particularly within the college population. Much of the literature review focuses on the nature of critical thinking, including its relationship with intelligence, its multifaceted character, and its components. However, given the study’s aim to investigate the development of critical thinking and emotional intelligence, as well as the relationship between them, it is essential to provide background information on the status of critical thinking in higher education.
Second, the descriptions of the instruments used are inadequate, particularly since both measures are lesser known and were not originally developed in English. For instance, the “critical thinking tasks” involve 15 open- and close-ended questions evaluated using a rubric scoring between 0 and 2 points. However, the study does not provide sample items, leaving readers unable to assess the structure of the survey or its face validity. Including example questions would significantly enhance the readers’ understanding.
Third, the results lack meaningful contributions to existing knowledge about either construct or their relationship. The results section is organized into three parts: descriptive statistics, correlations, and gender and age differences. However, none of these findings are particularly noteworthy. The interpretation of results is also problematic. For example, the authors assert that there is a "moderate level of critical thinking and a high level of emotional intelligence," but fail to specify the basis for these conclusions. Compared to what benchmarks? Similarly, the claim of gender differences is questionable; a p-value of .091 is not significant, especially with a large sample size, suggesting a negligible effect size.
The rationale for grouping participants into age categories—19–20, 21–23, and 24–47 years—is also unclear. A correlational analysis might have been more straightforward. Furthermore, the interpretation of higher critical thinking scores among 19–20-year-olds compared to older groups is unconvincing. This discrepancy could point to measurement validity issues or other artifacts, which cannot be adequately assessed due to insufficient details about the instruments.
Overall, while the study has some merit, its lack of a robust literature review, insufficient methodological detail, and weak results undermine its significance. Addressing these issues would enhance the study's contributions to the field.
Author Response
Comment 1: First, the study lacks a thorough introduction to the current state of research on critical thinking, particularly within the college population. Much of the literature review focuses on the nature of critical thinking, including its relationship with intelligence, its multifaceted character, and its components. However, given the study’s aim to investigate the development of critical thinking and emotional intelligence, as well as the relationship between them, it is essential to provide background information on the status of critical thinking in higher education.
Response 1: Paragraphs about research in college and university students was added in the introduction. The relationship between CT and EI and researches has been enlarged.
Comment 2: Second, the descriptions of the instruments used are inadequate, particularly since both measures are lesser known and were not originally developed in English. For instance, the “critical thinking tasks” involve 15 open- and close-ended questions evaluated using a rubric scoring between 0 and 2 points. However, the study does not provide sample items, leaving readers unable to assess the structure of the survey or its face validity. Including example questions would significantly enhance the readers’ understanding.
Response 2: A figure showing an example of one of the questions was incorporated
Comment 3: Third, the results lack meaningful contributions to existing knowledge about either construct or their relationship. The results section is organized into three parts: descriptive statistics, correlations, and gender and age differences. However, none of these findings are particularly noteworthy. The interpretation of results is also problematic. For example, the authors assert that there is a "moderate level of critical thinking and a high level of emotional intelligence," but fail to specify the basis for these conclusions. Compared to what benchmarks? Similarly, the claim of gender differences is questionable; a p-value of .091 is not significant, especially with a large sample size, suggesting a negligible effect size.
Response: The paragraph about results was revised and changed. In the gender differences, we point out the p value of 0,91 like no significant but only when researcher assume the standard rules of 95% of confidence, but there is a confidence level of 90%, weekly than other, but usable. We applied a covariance analysis precisely to confirm that non-significance founded.
Comment 3: The rationale for grouping participants into age categories—19–20, 21–23, and 24–47 years—is also unclear. A correlational analysis might have been more straightforward. Furthermore, the interpretation of higher critical thinking scores among 19–20-year-olds compared to older groups is unconvincing. This discrepancy could point to measurement validity issues or other artifacts, which cannot be adequately assessed due to insufficient details about the instruments.
Response 3: We included a major description about the CT questionnaire. The grouping by age is explained by belonging to different levels of education. This is detailed in the participants and in the results. The explanation about the differences between ages is a hypothesis that arises, but is relativized in the discussion.
Reviewer 3 Report
Comments and Suggestions for Authors
This paper is an interesting and potentially useful one, but there are several omissions that would improve the paper significantly if they were to be addressed:
1. the rationale for the paper is unclear and not well argued. It is very prescriptive (it uses the word should a lot) and does not give a clear and persuasive argument for its purpose.
2. in the literature proper, in each major section, there is a concentration on defining concepts, rather than explaining their purpose, and why they are increasingly important in the 21st-century world.
3. in the methodology, there are very details about the sample, how they were recruited, what their subjects are, and how ethical approval was sought fro each of them.
4. in the discussion, the literature was limited to quite technical issues and not to the purpose of the two major concepts, which in terms of the overall significance and transfer-ability of the study, are extremely important.
Author Response
Comment 1: 1. the rationale for the paper is unclear and not well argued. It is very prescriptive (it uses the word should a lot) and does not give a clear and persuasive argument for its purpose.
Response 1: An explanation about the purpose of the text was included in Introduction. Writing style was revised too.
Comment 2: in the literature proper, in each major section, there is a concentration on defining concepts, rather than explaining their purpose, and why they are increasingly important in the 21st-century world.
Response: The definition of concepts are relevant, the purpose of it is included in the discussion. The reference to XXI century skills, was diminished.
Comment 3: In the methodology, there are very details about the sample, how they were recruited, what their subjects are, and how ethical approval was sought fro each of them.
Response 3: Information in methodology was expanded, the desdription of participants was described more deeply. Ethical statements are described in the text.
Comment 4: in the discussion, the literature was limited to quite technical issues and not to the purpose of the two major concepts, which in terms of the overall significance and transfer-ability of the study, are extremely important.
Response: In the discussion ideas about two concepts are expanded, and a paragraph with ideas sobout how apply some actions to promote CT was included.
Reviewer 4 Report
Comments and Suggestions for Authors
This paper is probably better suited to a journal that doesn’t focus on intelligence research. I say this mainly because emotional intelligence and critical thinking have been argued to be construct redundant with the structure of cognitive abilities and other measurable traits.
That said, it’s okay to publish this paper in a journal focused on intelligence as long as you make these qualifications and limitations known so that at least the research is put in the proper context and might make a contribution to the field of intelligence – however, again, without controlling for these other constructs, it’s unclear what your study really shows.
Author Response
Comment 1: This paper is probably better suited to a journal that doesn’t focus on intelligence research. I say this mainly because emotional intelligence and critical thinking have been argued to be construct redundant with the structure of cognitive abilities and other measurable traits.
Response 1: The purpose to publish the paper in a journal of intelligence, is contribute to understanding of links between IQ and others processes both cognitive and non cognitive.
Comment 2: That said, it’s okay to publish this paper in a journal focused on intelligence as long as you make these qualifications and limitations known so that at least the research is put in the proper context and might make a contribution to the field of intelligence – however, again, without controlling for these other constructs, it’s unclear what your study really shows.
Response 2: The study seeks to show the relationship between both concepts in a group of future teachers, given the importance of both processes in Chilean educational policy. Although the data are weak, the discussion highlights the importance of continuing research in this line to obtain a greater amount of empirical information.
Reviewer 5 Report
Comments and Suggestions for Authors
The article is of interest and current relevance. However, it could be improved by incorporating some adjustments, detailed below:
-
Limitations in the generalization of results: The findings may not be applicable to other disciplines or social contexts outside the pedagogy programs studied at a Chilean university. It´s recommended to address this limitation more explicitly in the discussion or conclusions of the study. Although it´s mentioned, it isn´t sufficiently analyzed.
-
Review of the title: It´s suggested to adjust the title to reflect that the sample corresponds to a single institution, ensuring greater precision and alignment with the study's content.
-
Concrete measures to promote skills: While the need to implement measures to foster critical thinking and emotional intelligence among students is mentioned, the proposals are too general. It would be valuable to include more specific and detailed measures that could serve as practical guidelines for educational institutions.
These improvements would add clarity, depth, and applicability to the article, enhancing its impact and usefulness in both academic and practical contexts.
Author Response
Comment 1: Limitations in the generalization of results: The findings may not be applicable to other disciplines or social contexts outside the pedagogy programs studied at a Chilean university. It´s recommended to address this limitation more explicitly in the discussion or conclusions of the study. Although it´s mentioned, it isn´t sufficiently analyzed.
Response: A explicitation about non applicability of this results was described in te discussion.
Comment 2: Review of the title: It´s suggested to adjust the title to reflect that the sample corresponds to a single institution, ensuring greater precision and alignment with the study's content.
Response: The title was changed
Comment 3: Concrete measures to promote skills: While the need to implement measures to foster critical thinking and emotional intelligence among students is mentioned, the proposals are too general. It would be valuable to include more specific and detailed measures that could serve as practical guidelines for educational institutions.
Response: The actions that can be used to strengthen critical thinking in education were expanded in detail in the discussion.
Round 2
Reviewer 1 Report
Comments and Suggestions for Authors
This version of the paper is better: It now addresses whether the relationships might be spurious, with each caused by intelligence That said, to me this paper still is not quite there. Though Willingham now gets a mention, the paper needs more from the critics of both concepts. Using chapter 2 in David Steiner’s just published A Nation at Thought (Rowman & Littlefield, 2024) could help. These concepts are not very coherent, and it is unclear that training in them improves teaching in measurable ways.
Relatedly, critical thinking depends on knowledge; else we have nothing to be sensibly critical about. This should be explored more
Author Response
Comment 1:
Though Willingham now gets a mention, the paper needs more from the critics of both concepts. Using chapter 2 in David Steiner’s just published A Nation at Thought (Rowman & Littlefield, 2024) could help. These concepts are not very coherent, and it is unclear that training in them improves teaching in measurable ways.
Response:
We include in the introduction, two paragraphs analyzing this points, including the Steiner's point of view and counterarguing ideas about the fuzzy nature of two concepts, but using the way of studying them through the operationalization of variables used in measurements studies.
Comment2: Relatedly, critical thinking depends on knowledge; else we have nothing to be sensibly critical about. This should be explored more.
Response: We pointed out the critical tinking's perspective of C. Saiz, who derived his models of D. Halpern; they arguing that CT is a manner of use the thought, and knowledge can be a final step in this use. A paragraph in critical thinking section of introduction make a mention of this point.
Reviewer 2 Report
Comments and Suggestions for Authors
I command the authors making effort to address my questions especially in the literature review part. However, I am still having reservations regarding the findings:
(1) Unimpressive findings for almost all factors that investigated: no convincing statistics in terms of gender and age differences in EM and CT. I am not convinced by the so called "three-age group comparison." There is a big dip in the second age group (22-23 years old), in comparison between the other two age groups (20-21, and above 23). No one would buy the argument that the first two age groups (20-21) and (22-23) are in some sort of different generations! The significant dig in the second age group can be any unclear artifacts (e.g., sample errors). A more convincing analysis regarding age difference in either EM or CT should be correlation analysis. I am surprised that it was not even mentioned as it would make more sense than the forced age group breakdown ANOVA.
(2) I don't really get how the authors made the claim that their participants received moderately higher level of EM and CT? There is no benchmark number (aside from average) for such a claim. Any sample can have higher than average numbers (assuming there is an established population mean for the measure), does that qualify it as moderately high? Without any evidence from inferential statistics, any number is subjective!
The only merit of the finding is the significant correlation between EM and CT. Again, the number is rather unconvincing (r-.2)
Author Response
Comments 1: (1) Unimpressive findings for almost all factors that investigated: no convincing statistics in terms of gender and age differences in EM and CT. I am not convinced by the so called "three-age group comparison." There is a big dip in the second age group (22-23 years old), in comparison between the other two age groups (20-21, and above 23). No one would buy the argument that the first two age groups (20-21) and (22-23) are in some sort of different generations! The significant dig in the second age group can be any unclear artifacts (e.g., sample errors). A more convincing analysis regarding age difference in either EM or CT should be correlation analysis. I am surprised that it was not even mentioned as it would make more sense than the forced age group breakdown ANOVA.
Response 1: We think it is a logic way to analyze the gender and age differences by mean's compared groups, because anothers studies had same approach.
Comment 2: I don't really get how the authors made the claim that their participants received moderately higher level of EM and CT? There is no benchmark number (aside from average) for such a claim. Any sample can have higher than average numbers (assuming there is an established population mean for the measure), does that qualify it as moderately high? Without any evidence from inferential statistics, any number is subjective!
Response 2: in the first paragrpah of results we correct the sentence, and now it is only descriptive. But we think it is posible analyze (lightly) the values observed in the mean of variables, in compared to the reported values ​​of the minimum and maximum scores of the scales.
Comment 3: The only merit of the finding is the significant correlation between EM and CT. Again, the number is rather unconvincing (r-.2)
Response 3: the data shows this values, and we consider it weak, and it is considered in 3rd paragraph in Discussion section. But we pointed out the importance of deepening the findings to continue generating knowledge on this topic that has been little studied empirically
Reviewer 3 Report
Comments and Suggestions for Authors
Thankyou for addressing my comments, which you have carried out carefully and thoroughly.
Author Response
Not new comments. We appreciate the suggestions given previously
Reviewer 4 Report
Comments and Suggestions for Authors
The responses to my comments are inadequate.
Author Response
Comment 1:
This paper is probably better suited to a journal that doesn’t focus on intelligence research. I say this mainly because emotional intelligence and critical thinking have been argued to be construct redundant with the structure of cognitive abilities and other measurable traits.
Response 1: We pointed out this study is based on perspectives that define critical thinking and emotional intelligence, like concepts operationalized in measurement scales, and used in a different researches. This is a way to work with fuzzy cognitive process.
Comment 2: That said, it’s okay to publish this paper in a journal focused on intelligence as long as you make these qualifications and limitations known so that at least the research is put in the proper context and might make a contribution to the field of intelligence – however, again, without controlling for these other constructs, it’s unclear what your study really shows.
Response 2: this study has a correlational design, because the relation between critical thinking and emotiional intelligence has some few empirical studies; From our point of view, it is necessary to contribute with more studies to better understand this topic.
Reviewer 5 Report
Comments and Suggestions for Authors
Dear authors,
With the revisions made to your work, its quality has significantly improved, and it is now ready for publication.
Best regards.
Author Response

(The authors gave the same response as above.)
